# Distinct Subtyping of Successful Weaning from Acute Kidney Injury Requiring Renal Replacement Therapy by Consensus Clustering in Critically Ill Patients

**DOI:** 10.3390/biomedicines10071628

**Published:** 2022-07-07

**Authors:** Heng-Chih Pan, Chiao-Yin Sun, Thomas Tao-Min Huang, Chun-Te Huang, Chun-Hao Tsao, Chien-Heng Lai, Yung-Ming Chen, Vin-Cent Wu

**Affiliations:** 1Graduate Institute of Clinical Medicine, College of Medicine, National Taiwan University, Taipei 100, Taiwan; hengchihpan0107@gmail.com; 2Division of Nephrology, Department of Internal Medicine, Community Medicine Research Center, Keelung Chang Gung Memorial Hospital, Keelung 204, Taiwan; fish3970@gmail.com; 3School of Medicine, Chang Gung University College of Medicine, Taoyuan 33302, Taiwan; 4Division of Nephrology, Department of Internal Medicine, National Taiwan University Hospital, Taipei 100, Taiwan; taomin.huang@gmail.com (T.T.-M.H.); chenym@ntuh.gov.tw (Y.-M.C.); 5Nephrology and Critical Care Medicine, Department of Internal Medicine and Critical Care Medicine, Taichung Veterans General Hospital, Taichung 407, Taiwan; huangchunte@gmail.com; 6Department of Surgery, National Taiwan University Hospital, Taipei 100, Taiwan; zeus360801@gmail.com (C.-H.T.); b8805007@gmail.com (C.-H.L.)

**Keywords:** acute kidney injury, clustering algorithm, critically ill patient, dialysis-free, mortality, renal replacement therapy

## Abstract

**Background:** Clinical decisions regarding the appropriate timing of weaning off renal replacement therapy (RRT) in critically ill patients are complex and multifactorial. The aim of the current study was to identify which critical patients with acute kidney injury (AKI) may be more likely to be successfully weaned off RRT using consensus cluster analysis. **Methods:** In this study, critically ill patients who received RRT at three multicenter referral hospitals at several timepoints from August 2016 to July 2018 were enrolled. An unsupervised consensus clustering algorithm was used to identify distinct phenotypes. The outcomes of interest were the ability to wean off RTT and 90-day mortality. **Results:** A total of 124 patients with AKI requiring RRT (AKI-RRT) were enrolled. The 90-day mortality rate was 30.7% (38/124), and 49.2% (61/124) of the patients were successfully weaned off RRT for over 90 days. The consensus clustering algorithm identified three clusters from a total of 45 features. The three clusters had distinct features and could be separated according to the combination of urinary neutrophil gelatinase-associated lipocalin to creatinine ratio (uNGAL/Cr), Sequential Organ Failure Assessment (SOFA) score, and estimated glomerular filtration rate at the time of weaning off RRT. uNGAL/Cr (hazard ratio [HR] 2.43, 95% confidence interval [CI]: 1.36–4.33) and clustering phenotype (cluster 1 vs. 3, HR 2.7, 95% CI: 1.11–6.57; cluster 2 vs. 3, HR 44.5, 95% CI: 11.92–166.39) could predict 90-day mortality or re-dialysis. **Conclusions:** Almost half of the critical patients with AKI-RRT could wean off dialysis for over 90 days. Urinary NGAL/Cr and distinct clustering phenotypes could predict 90-day mortality or re-dialysis.

## 1. Background

Acute kidney injury (AKI) is a common syndrome that has a significant impact on patient prognosis in various clinical settings. Patients with AKI requiring renal replacement therapy (AKI-RRT) are considered to be critically ill, are estimated to account for up to 60% of patients in intensive care units (ICUs), and have a high mortality rate ranging from 38% to 80% [1,2,3]. Successful weaning off RRT is associated with a decreased ICU length of stay and favorable prognosis [4,5,6]. Currently, physicians usually make the decision to wean off RRT based on criteria including clinical status, timed urine creatinine clearance, and urine output [7]. Nevertheless, these parameters can be affected by multiple factors, some of which do not directly reflect renal repair and prognosis [6].

A data-driven deep learning approach has been used to identify subtypes of AKI with different outcomes and response to therapy on the basis of the results of a set of unstructured data [8,9]. It has been shown to be especially valuable to investigate medical diseases with high heterogeneity, such as chronic kidney disease [10] and AKI [9].

However, the utility of data-driven deep learning to investigate heterogenous distribution patterns of AKI-RRT patients attempting to wean off dialysis remains unknown. Furthermore, emerging evidence has shown that several biomarkers are sensitive, specific, and causally related with regards to the early prediction or risk assessment of AKI, including neutrophil gelatinase-associated lipocalin (NGAL) and liver-type fatty-acid binding protein (L-FABP) [11,12,13,14,15]. Nevertheless, it is unclear whether these biomarkers can be used to distinguish the patients who will remain dialysis-dependent after attempting to wean off RRT from those who will be successfully weaned and remain off RRT.

We hypothesized that a data-driven deep learning method incorporating clinical data and the latest biomarkers could more accurately identify the risk factors and predict which AKI-RRT patients could be successfully weaned off RRT. Therefore, the aim of this study was to determine whether unsupervised deep-learning-based consensus clustering could identify clusters of critically ill AKI-RRT patients attempting to wean off RRT and investigate different distribution patterns of mortality and re-dialysis among them.

## 2. Methods

### 2.1. Study Design and Population

This study was conducted using a prospectively created AKI database with patients from National Taiwan University Hospital, Chang Gung Memorial Hospital, and Taichung Veterans General Hospital from August 2016 to July 2018 [16,17,18,19,20,21,22]. We enrolled critically ill adult patients with AKI-RRT who met the following criteria: (1) those whose intrinsic renal function had adequately recovered [23]; (2) RRT was no longer consistent with the treatment goals [23]; (3) the indication for starting RRT (azotemia with overt uremic symptoms, refractory hyperkalemia, oliguria or anuria refractory to diuretics, fluid overload refractory to diuretics along with pulmonary edema, severe metabolic acidosis) was in remission; and (4) a trend toward decreasing serum creatinine (sCr), urine output ≥400 mL/24 h with or without diuretics, and improved fluid overload, electrolyte and metabolic status [7,17]. The exclusion criteria were: (1) age < 18 years; (2) previous nephrectomy, renal transplantation or RRT treatment; (3) ICU or hospital length of stay of respectively <2 days and >180 days during the index hospitalization; (4) patients who resumed dialysis within 48 h because of a new onset of a critical episode; and (5) patients with AKI caused by surgically induced injury, vasculitis, obstruction, glomerulonephritis, interstitial nephritis, hemolytic uremic syndrome, or thrombotic thrombocytopenic purpura (the study design is provided in the Appendix A).

### 2.2. Data Collection and Definitions of Variables

#### 2.2.1. Baseline Measurements and Characteristics

The prospectively collected variables in the database included demographic characteristics, etiology of AKI, indications for initial dialysis and re-dialysis, and disease severity according to Sequential Organ Failure Assessment (SOFA) score, which was calculated at (1) 24 h before dialysis initiation (T2), (2) the time of weaning off dialysis (T3), and (3) 24 h after weaning off dialysis (T4). Blood and urine samples were also collected at the time of disease severity evaluation and baseline. Baseline (T1) SCr was defined as the nadir value during the last hospitalization within the last 365 days, or the mean SCr value more than 180 days before the index admission in those without a previous admission [24,25].

#### 2.2.2. Indication for Renal Replacement Therapy

The RRT modality in each patient was initially chosen by the attending physician and adjusted accordingly based on disease evolution by a critical care nephrologist. The indication for RRT included one or more of the following: (1) azotemia (blood urea nitrogen (BUN) > 80 mg/dL and sCr > 2 mg/dL) with overt uremic symptoms (encephalopathy, pericarditis, or pleuritis); (2) refractory hyperkalemia with serum potassium level > 5.5 mmol/L; (3) oliguria (urine output < 400 mL/24 h) or anuria (<100 mL/24 h) refractory to diuretics; (4) fluid overload refractory to diuretics along with a central venous pressure > 12 mmHg or pulmonary edema with PaO2/FiO2 < 300 mmHg; and (5) metabolic acidosis (pH < 7.2 in arterial blood) [16,26].

#### 2.2.3. Measurement of Urinary Biomarker Levels

At the time of stopping RRT, urine samples were collected and stored at −80 °C until analysis. Urinary NGAL (uNGAL) and urinary L-FABP (uL-FABP) levels were determined using enzyme-linked immunosorbent assay (ELISA) kits (NGAL, R&D, Minnesota, USA; L-FABP, Sekisui Medical Co., Ltd., Tokyo, Japan). Each biomarker assay was performed in duplicate according to the manufacturers’ instructions, and the mean value was used for further statistical analysis (the detection ranges of the kits are provided in the Appendix A).

#### 2.2.4. Outcome Assessment

The primary outcomes were: (1) 90-day mortality after hospital discharge, and (2) the composite outcome of 90-day mortality and remaining on dialysis. All patients were followed until death or at least 90 days after discharge from the index hospitalization, whichever occurred first. Successful weaning off RRT was defined as survival without dialysis at the end of the study. Patients who received dialysis and planned to receive palliative care were excluded.

#### 2.2.5. Statistical Analysis

Continuous data were expressed as mean ± standard deviation, and categorical data were expressed as number (percentage). All variables were tested for normal distribution using the Kolmogorov–Smirnov test. The Student’s *t*-test was used to compare the means of continuous variables and normally distributed data; otherwise, the Mann–Whitney U test was used. The χ^2^ test or Fisher’s exact test was used to compare categorical data. We performed consensus clustering analysis on all of the study participants. The clustering algorithm allowed us to determine what number of clusters, k, best fit the data. Unsupervised hierarchical clustering of the selected variables was performed, and the optimal number of clusters was determined using a consensus matrix heatmap and consensus cumulative distribution function (CDF) [27]. The standardized mean differences of the input variables between each cluster and the overall study population were calculated and presented graphically to examine the cluster profiles. The data matrix was visualized using heat maps and circular plots, which allowed us to simultaneously visualize clusters of samples and the selected variables [28]. Generalized pair plots were used to visualize paired combinations of categorical and quantitative variables [29]. We compared the distribution of 18 selected variables across clusters using analysis of variance (ANOVA) and the χ^2^ test. We then calculated and graphically presented the standardized mean differences of the 18 selected variables across clusters and the overall study population to examine the cluster profiles. Net reclassification improvement (NRI) and integrated discrimination improvement (IDI) analyses were used to examine the role of consensus clustering and uNGAL/Cr and stratify individuals into higher or lower risk categories (re-classification) [14,25,30].

Multivariate Cox proportional hazards analysis was carried out on selected variables that significantly affected the overall survival in univariate analysis and those identified in a literature review. The *p*-values less than 0.05 were considered to indicate statistical significance, and relationships between clusters and significant variables were displayed using 3D scatter plots. The main analysis of three clusters and the primary outcomes was conducted using the Kaplan–Meier method to plot cumulative survival curves, which were compared using the log-rank test. All analyses were performed using R software, version 3.2.2 (Free Software Foundation Inc., Boston, MA, USA).

## 3. Results

### 3.1. Study Population Characteristics

A total of 124 adult patients with AKI-RRT who fulfilled the inclusion criteria were enrolled in this study. The characteristics of the patients are listed in Table 1. The overall 90-day mortality rate was 30.7% (38/124), and 49.2% (61/124) of the patients were successfully weaned off dialysis for over 90 days (dialysis-free). The mean age of the study population was 61.7 years, and 35 patients (28.2%) were women. Among all participants, 55 (44.4%) had diabetes mellitus and 66 (53.2%) had hypertension. The overall mean baseline creatinine (Cr) and estimated glomerular filtration rate (eGFR) were 1.89 mg/dL and 64.58 mL/min per 1.73 m^2^, respectively. AKI could be attributed to shock in 93 patients (75.0%) and sepsis in 61 patients (49.2%). The overall mean SOFA score at dialysis initiation was 10.7 ± 3.9, and the quick sequential organ failure assessment (qSOFA) score at dialysis initiation was 1.0 ± 0.8. The leading reason for RRT was oliguria/anuria (53.2%), followed by fluid overload (42.7%). The main reason for re-dialysis was azotemia (26.6%), followed by oliguria/anuria (12.1%).

### 3.2. Unsupervised Cluster Analysis to Identify AKI Clusters

The consensus clustering algorithm identified 18 variables from a total of 45 features (Table 1). Using transformed values of these combined features, k-means clustering ranged from K = 2 to K = 6. The consensus matrix heatmaps for each cluster size, showing the pairwise consensus of all participants, are shown in Figure 1A–D. Figure 1E,F show the curves of cumulative distribution, and illustrate that the results were stable when the number of clusters (K) was three. Therefore, three clusters represented the best performance of the data pattern of our AKI-RRT population who could wean off dialysis. Cluster 1 had 30 (24.2%) patients, cluster 2 had 16 (12.9%) patients, and cluster 3 had 78 (62.9%) patients.

### 3.3. Clinical Characteristics of the Distinct Clusters

Table 1 lists the characteristics of the three clusters. The clinical features differed significantly among the three clusters. However, there were no significant differences in age, gender, baseline eGFR, and the prevalence of diabetes, AKI due to shock, indication for dialysis and clinical parameters after being weaned off RRT for 24 h (Table 1). Figure 2A shows the heatmap of the selected variables by unsupervised hierarchical clustering which separated all patients into three clusters. Figure 2B illustrates the matrix correlation of the top 12 features among the three clusters. The highest correlation coefficients among the three clusters were found between eGFR at baseline (T1) and when weaning off RRT (T3) (r = 0.867, *p* < 0.01), SCr and BUN before initiating RRT (T2) (r = 0.544, *p* < 0.01), and SCr before initiating RRT (T2) and after weaning off RRT for 24 h (T4) (r = 0.622, *p* < 0.01). These findings suggested that the renal function parameters of the patients were well correlated at the different timepoints of the study.

Circular barplots showing the selected predictors in the three clusters are depicted in Figure 3A. The discrepancies of most factors were within one standard difference of the three clusters. Standardized difference plots were used to visualize the key predictors of each cluster. Variables with an absolute standardized difference >0.3 were marked as the key features for each cluster (Figure 3B). Cluster 1 included patients with a relatively high severity of critical illness before initiating RRT (T2) and the highest uNGAL/Cr when weaning off RRT (T3). Cluster 2 included patients with relatively good renal function at all four time points as well as the highest severity of critical illness before initiating RRT (T2) and when weaning off RRT (T3). Cluster 3 included patients with the worst renal function at baseline (T1) and when weaning off RRT (T3).

### 3.4. Etiologies of AKI and Dialysis

In this study, the etiology of AKI was mostly attributed to shock (75.0%), including 23 of 30 (76.7%) patients in cluster 1, 10 of 16 (62.5%) patients in cluster 2, and 60 of 78 (76.9%) patients in cluster 3. There was a statistically significant difference in the prevalence of sepsis, with the lowest proportion in cluster 3. The indication for dialysis was mostly attributed to oliguria/anuria, followed by fluid overload. Cluster 2 had a significantly lower proportion of oliguria/anuria than the other two clusters (Table 1). Appendix A shows the reasons for re-dialysis. The indication for re-dialysis was mostly attributed to azotemia. Cluster 1 had a significantly higher proportion of oliguria/anuria, while cluster 2 had a significantly higher proportion of uremic symptoms.

### 3.5. AKI Phenotypes Predicting Clinical Outcomes

Overall, 20.2% (25/124) of the patients and 29.1% (25/86) of the survivors were dialysis-dependent at 90 days after attempting to wean off dialysis. Kaplan–Meier survival plots disclosed that each cluster phenotype was significantly associated with 90-day mortality (log rank *p* < 0.001, Figure 4A) as well as the composite outcome of 90-day mortality or re-dialysis (log rank *p* < 0.001, Figure 4B). Clusters 3, 1, and 2 were associated with a low, medium, and high risk of the clinical outcomes, respectively. Cox proportional hazards analysis showed that age (hazard ratio [HR] 1.06, 95% confidence interval [CI]: 1.02–1.10), uNGAL/Cr (HR 3.68, 95% CI: 1.63–8.31), eGFR when weaning off RRT (HR 0.96, 95% CI: 0.94–0.99), and acid-base imbalance (HR 2.59, 95% CI: 1.12–5.98) as an indication for dialysis were independent predictors of 90-day mortality. Clusters 1 and 2 had significantly higher HRs of mortality than cluster 3 (cluster 1 vs. 3: HR 3.69, 95% CI: 1.25–10.93; cluster 2 vs. 3: HR 26.2, 95% CI: 5.42–126.65) (Table 2). uNGAL/Cr was also an independent factor of the composite outcome of mortality or re-dialysis (HR 2.43, 95% CI: 1.36–4.33). Cluster 1 (HR 2.7, 95% CI: 1.11–6.57) and cluster 2 (HR 44.5, 95% CI: 11.92–166.39) still had significantly higher HRs than cluster 3 (Table 3). Figure 5 shows that a combination of uNGAL/Cr, SOFA, and eGFR at weaning off dialysis could significantly delineate the distinct pattern of the clusters and separate them from one another. Appendix A demonstrate that a combination of uNGAL/Cr and clustering phenotype could significantly delineate the distinct pattern of 90-day mortality. Furthermore, NRI and IDI analyses were used to distinguish risk categories and reclassify the patients who died from all causes at 90 days after RRT initiation into high and low risk categories. Combining uNGAL/Cr with clustering phenotype at the time of weaning off RRT led to a significant increase in risk stratification (categorical NRI = 0.336; 95% CI: 0.092–0.580; *p* = 0.007). Most of this effect came from the survivors (event IDI = 0.289; 95% CI: 0.084–0.495; *p* = 0.006). Likewise, the total IDI was significant (0.168; 95% CI: 0.045–0.291; *p* = 0.008).

## 4. Discussion

In this study, we used an unsupervised consensus clustering algorithm with 45 variables and identified three AKI-RRT clusters with different risks of mortality or re-dialysis after weaning off RRT. The 90-day mortality rate was 30.7% (38/124), and 49.2% (61/124) of the patients were successfully weaned off RRT for over 90 days, which is consistent with previous reports [6,31]. Moreover, the three clusters could be distinguished by incorporating SOFA score, eGFR, and uNGAL/Cr when weaning off RRT. Of note, distinct clustering phenotype and uNGAL/Cr were associated with 90-day mortality or re-dialysis.

The consensus clustering algorithm identified three clusters from 18 selected features which could represent the diverse clinical data and separate the patients into groups by significant different entities. Moreover, the clusters were strongly associated with clinical outcomes. Despite differences in baseline renal function between the clusters, all three clusters met the Kidney Disease Improving Global Outcomes (KDIGO) stage 3 AKI. The cluster phenotypes provided a simple metric summarizing the heterogeneity of critical illness and clinical presentation.

Of note, the patients in the worst cluster (cluster 2) had the best renal function parameters, including higher baseline eGFR, lower BUN and SCr before initiating RRT, higher eGFR and bicarbonate level when weaning off RRT, as well as lower BUN and SCr at 24 h after weaning off RRT. These features have been associated with a good prognosis [17,26,32]. However, the patients in cluster 2 also had more severe illness in terms of higher qSOFA score and lower bicarbonate level before initiating RRT, which have been associated with a poor prognosis [26,33]. The results showed that clustering analysis could cluster baseline characteristics with a temporal change in disease severity into distinct phenotypes.

Consensus clustering is a more robust approach which relies on multiple iterations of the chosen clustering method on sub-samples of the dataset. Using this method, we could identify the potential heterogeneity factors linked to the outcomes, and build a putative model to distinguish patients who were at a higher risk [10]. Notably, we found a strong independent association between the cluster phenotype and adverse outcomes after controlling for other independent variables measured at different timepoints.

Because a considerable number of hospitalized patients develop AKI-RRT in a wide variety of heterogeneous clinical settings, consensus clustering analysis may allow for better prediction of the prognosis along with objective information for clinical decision-making when treating this subset of patients. In this study, cluster 3 had the worst kidney function (the lowest eGFR) while cluster 2 had the best kidney function (the highest eGFR and bicarbonate level) when weaning off RRT. According to current criteria and clinical practice, the patients in cluster 3 may be thought to have the lowest probability and patients in cluster 2 have the highest probability of successful weaning off RRT, in contrast to our findings. These results highlight the importance of incorporating disease severity and baseline kidney function to accurately predict patient outcomes.

NGAL is a widely expressed 25 kDa protein belonging to the lipocalin family, which transports small hydrophobic molecules such as steroids and lipids [34]. Urinary NGAL is produced in renal epithelia and leukocytes in response to tubular injury and systemic inflammation, and it can be detected as early as 3 h after nephrotoxic or ischemic injury [35,36,37,38]. The prognostic value of uNGAL and uNGAL/Cr for kidney recovery has been studied [31,39,40,41]. Srisawat et al. reported that lower uNGAL levels during the first 14 days following AKI were associated with a reduced need for renal support at 60 days in a setting of AKI-RRT [31]. One recent retrospective study by Lumlertgul et al. [42]. demonstrated that uNGAL/Cr showed the best discrimination ability for persistent AKI versus transient AKI, comparable to uNGAL alone. In the current study, our data further demonstrated that uNGAL/Cr was an important independent factor of 90-day mortality in AKI-RRT patients attempting to wean off RRT. Of note, it was the only independent predictor of the composite endpoint of mortality or re-dialysis. The NRI and IDI analyses also demonstrated that the integration of uNGAL/Cr could enhance the identification of patients who could be successfully weaned off RRT. Furthermore, cluster analysis also showed that the three distinct phenotypes could be separated according to the combination of uNGAL/Cr, SOFA score, and eGFR at the time of weaning off RRT (Figure 5), which also highlights the clinical impact of uNGAL/Cr on the heterogeneity in these patients.

To the best of our knowledge, this is the first study to apply consensus clustering analysis to investigate the potential risk factors or predictors of the outcomes of patients with AKI-RRT attempting to wean off dialysis. Our findings provide another aspect of grouping patients with similar characteristics who could have similar clinical outcomes. Using a data-driven approach, consensus clustering analysis identified three clusters of AKI-RRT patients on the basis of 45 variables, including baseline characteristics, comorbidities, clinical parameters and novel biomarker data (uNGAL/Cr and uL-FABP/Cr). Our results demonstrated that the three clusters were associated with different prognoses in the critically ill patients with AKI-RRT attempting to wean off RRT. The more comprehensively and precisely clinicians can determine which patient’s kidney function will recover and to what degree and how long it might take, the more the medical team will be able to individualize therapeutic strategies and care plans, and prepare patients for future developments. However, further studies are needed to investigate whether this approach could guide treatment or improve outcomes [10].

In spite of the encouraging results observed in this study, several potential limitations should be recognized. First, clinical decisions regarding when to wean off RRT were not based on standardized criteria; rather, decisions were made by the attending physicians according to their clinical judgment. However, this study was conducted at three tertiary care referral hospitals that have historically cooperated for over 10 years and collaborated on many multi-center research studies [16,17,18,19,20,21,22]. This long-term collaboration may have partially reduced the heterogeneity in clinical practice styles with regards to ceasing and re-initiating RRT. Second, there was limited information on suspected causes of sepsis and shock. In critical settings, multifactorial conditions could relate to sepsis- or shock-related AKI. Although it is a challenge to delineate the etiologies of shock or sepsis, the main purpose of this study was to investigate the indicators for patients who could wean from dialysis requiring AKI, namely recovery from shock or sepsis. While it is not perfect, our work is the first study to determine the likelihood of predicting 90-day mortality or re-dialysis. Third, sequential measurements of urinary biomarkers may reflect the dynamic aspects of clinical diseases, thus providing superior information on the risks of re-dialysis and mortality. Future studies evaluating other novel stress and tubular damage biomarkers using serial measurements may be more helpful in accurately evaluating kidney function recovery in AKI-RRT patients. Fourth, the relatively small sample size may have affected the statistical power. In addition, we separated our patients into three clusters, and the number of patients in each cluster varied. However, our analysis confirmed the optimal number of clusters by calculating the CDF for each consensus matrix from each cluster number, and discrete characteristics of each group were noted. Fifth, cluster analysis is a category of exploratory data analysis, and the data-driven approach relied on the input of data. The algorithm identified clusters of individuals who shared the same or similar characteristics in terms of the input variables. Sixth, information was lost due to the clustering approach summarizing the individual heterogeneity to provide better clinical interpretability. Lastly, the clustering approach was conducted on dialysis initiation to allow this study to be applied in clinical practice. Although the clusters were analyzed at multiple centers, future studies should include a validation dataset.

## 5. Conclusions

In summary, the three clusters of AKI-RRT patients who attempted to wean off RRT had discrete features and were highly associated with mortality and re-dialysis. uNGAL/Cr could be combined with SOFA score and eGFR at the time of weaning off RRT to separate the clusters from one another. Our results showed that consensus clustering analysis could improve the prediction of prognosis along with objective information for clinical decision-making when treating critical dialysis patients.

## Figures and Tables

**Figure 1 biomedicines-10-01628-f001:**
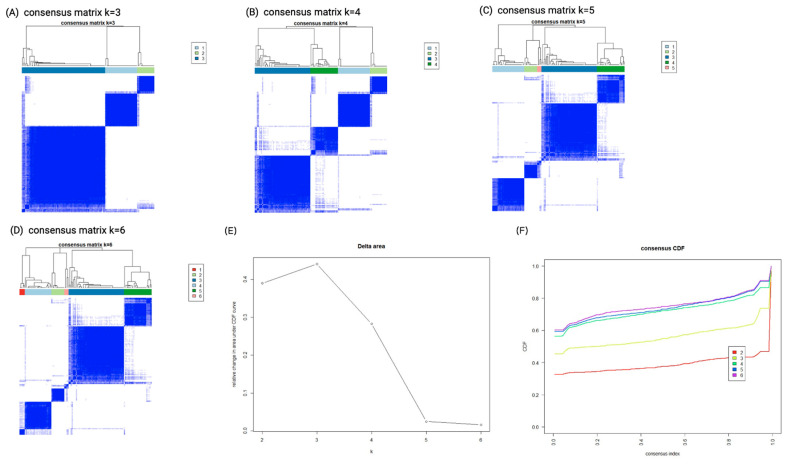
Consensus clustering analysis results displaying the robustness of sample classification using multiple iterations of k-means clustering. (**A**–**D**) The consensus matrix heat maps of K = 3, 4, 5 to K = 6 using 45 baseline parameters. The dark blue color shows the consensus of which groups fit perfectly together, and the white color shows that two individuals are always grouped separately. (**E**) Curve of CDF. (**F**) CDF delta area curve of consensus clustering. The *x* axis represents the category k, and the y axis denotes the relative change in area under the CDF curve of category k compared with category k − 1. **Abbreviations:** CDF, cumulative distribution function.

**Figure 2 biomedicines-10-01628-f002:**
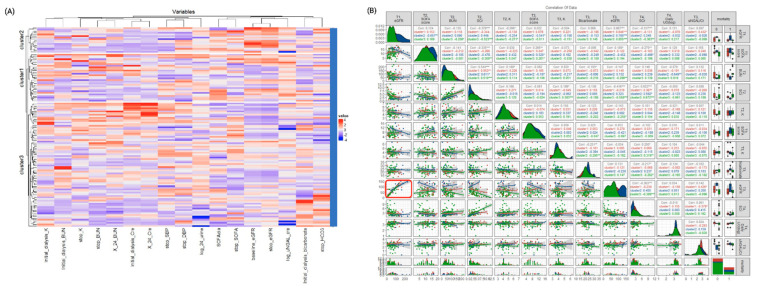
(**A**) **The heatmap shows the three subtypes according to the selected baseline parameters.** The dendrogram on the y axis shows consensus clustering of the patients. The x axis shows the variables of the patients attempting to wean from RRT. (**B**) **Correlations between baseline predictors with overall mortality.** Generalized pairs plot depicting all of the pairwise scatter plots comparing scores of each pair of selected predictors (upper diagonal), and the regression line (lower diagonal). Each plotted point represents a study participant. Distributions and plots are based on values for the area under the positive part of the curve which are displayed on the axes; the shaded areas show 95% confidence intervals, and significant differences are represented by asterisks. Red boxes represent significant correlations of baseline eGFR and eGFR when stopping dialysis. **Abbreviations:** BUN, blood urea nitrogen; Cr, Creatinine; eGFR, estimated glomerular filtration rate; NGAL, neutrophil gelatinase-associated lipocalin; RRT, renal replacement therapy; sCr, serum creatinine; SOFA, Sequential Organ Failure Assessment; UO, urine output.

**Figure 3 biomedicines-10-01628-f003:**
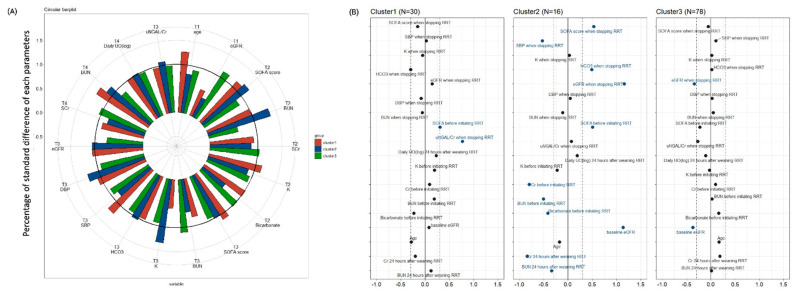
(**A**) **Circular barplots of the selected predictors associated with each cluster.** The height of each bar represents each percentage of standard difference to total standard difference response of each characteristic. (**B**) **Three clusters based on 18 parameters**. Manhattan plot of the standardized differences across three AKI subgroups of the selected baseline predictors. The x axis is the standardized difference value, and the y axis represents the four time serial baseline parameters. The dashed-dotted vertical lines represent standardized difference cutoffs of >0.3 or <−0.3. The light blue horizontal lines sort the category to which the predictors belong, including T1 (baseline), T2 (before initiating RRT), T3 (at the time of stopping RRT) and T4 (24 h after weaning off RRT). **Abbreviations:** AKI, acute kidney injury; BUN, blood urea nitrogen; Cr, creatinine; eGFR, estimated glomerular filtration rate; NGAL, neutrophil gelatinase-associated lipocalin; RRT, renal replacement therapy; sCr, serum creatinine; SOFA, Sequential Organ Failure Assessment; UO, urine output.

**Figure 4 biomedicines-10-01628-f004:**
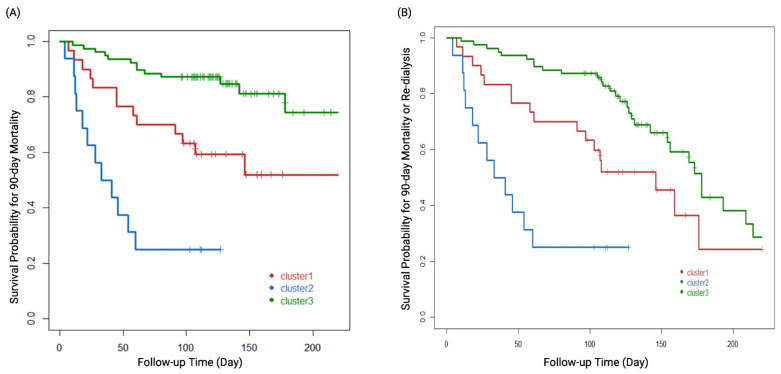
Kaplan–Meier survival plots of the three cluster groups defined by all predictors and (**A**) 90-day mortality. (**B**) Composite outcome of 90-day mortality or re-dialysis. The log rank *p* values for all comparisons were <0.001.

**Figure 5 biomedicines-10-01628-f005:**
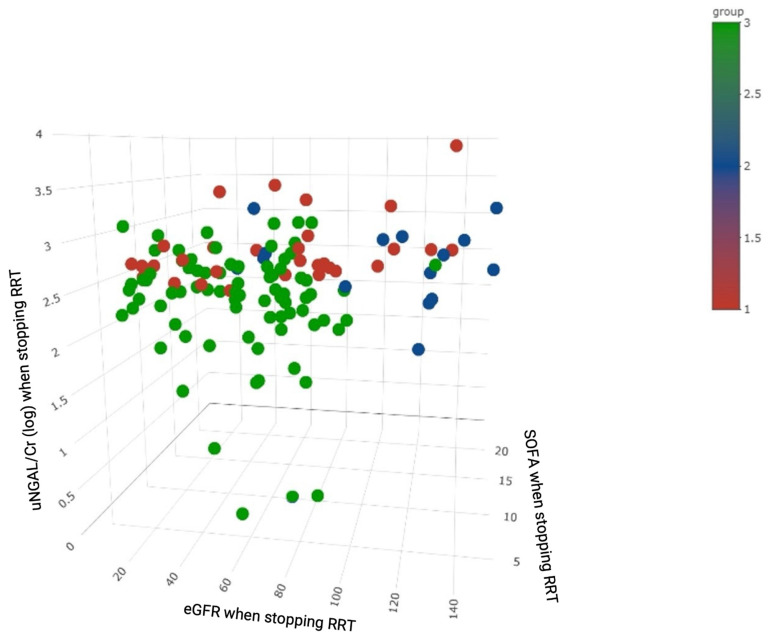
3D scatter plot of patients with colors representing clustering by eGFR, SOFA score and uNGAL/Cr (log) at the time of weaning off RRT. **Abbreviations:** Cr, creatinine; eGFR, estimated glomerular filtration rate; RRT, renal replacement therapy; SOFA, Sequential Organ Failure Assessment; uNGAL, urinary neutrophil gelatinase-associated lipocalin.

**Table 1 biomedicines-10-01628-t001:** Characteristics of enrolled patients.

Predictors	Total (*n* = 124)	Cluster ^1^ (*n* = 30)	Cluster ^2^ (*n* = 16)	Cluster ^3^ (*n* = 78)	*p* Value
**Demographic factors (T1)**					
Age, years	61.7 ± 16.7	56.6 ± 19.0	58.4 ± 20.2	64.4 ± 14.5	0.83
Gender (male), *n* (%)	89 (71.8%)	18 (60.0%)	10 (62.5%)	61 (78.2%)	0.11
Baseline SCr, mg/dL	1.9 ± 1.8	2.0 ± 2.2	0.8 ± 0.4	2.1 ± 1.7	**<0.001**
Baseline eGFR, mL/min/1.73 m^2^	64.6 ± 44.5	68.1 ± 44.1	124.9 ± 60.5	50.9 ± 27.7	0.19
Diabetic mellitus, *n* (%)	55 (44.4%)	14 (46.7%)	6 (37.5%)	35 (44.9%)	0.71
Hypertension, *n* (%)	66 (53.2%)	16 (53.3%)	7 (43.8%)	43 (55.1%)	**0.025**
Cardiorenal syndrome, *n* (%)	66 (53.2%)	10 (33.3%)	8 (50.0%)	48 (61.5%)	**0.030**
Type 1	55 (44.4%)	10 (33.3%)	8 (50.0%)	37 (47.4%)	**0.027**
Type 2	11 (8.9%)	0 (0.0%)	0 (0.0%)	11 (14.1%)	**<0.001**
Mechanical ventilator use	94 (75.8%)	26 (86.7%)	13 (81.3%)	55 (70.5%)	**0.017**
Infection	93 (75.0%)	25 (83.3%)	14 (87.5%)	54 (69.2%)	0.15
Time from diagnosis to RRT	10.8 ± 35.6	17.9 ± 53.9	10.7 ± 19.4	8.1 ± 29.0	0.51
**Etiology of AKI**					
AKI due to shock, *n* (%)	93 (75.0%)	23 (76.7%)	10 (62.5%)	60 (76.9%)	0.47
AKI due to sepsis, *n* (%)	61 (49.2%)	22 (73.3%)	9 (56.3%)	30 (38.5%)	**0.004**
AKI due to drug, *n* (%)	7 (5.6%)	2 (6.7%)	1 (6.3%)	4 (5.1%)	0.87
AKI due to contrast, *n* (%)	13 (10.5%)	3 (10.0%)	2 (12.5%)	8 (10.3%)	0.91
AKI due to all other cause *, *n* (%)	17 (13.7%)	4 (13.3%)	3 (18.8%)	10 (12.8%)	0.80
**Etiology of Shock**					
Septic shock, *n* (%)	18 (14.5%)	6 (20.0%)	4 (25.0%)	8 (10.3%)	0.16
Cardiogenic shock, *n* (%)	17 (13.7%)	2 (6.7%)	3 (18.8%)	12 (15.4%)	0.40
Hypovolemic shock, *n* (%)	2 (1.6%)	2 (6.7%)	0 (0.0%)	0 (0.0%)	0.073
**Indication for dialysis**					
Azotemia, *n* (%)	46 (37.1%)	12 (40.0%)	5 (31.3%)	29 (37.2%)	0.44
Fluid overload, *n* (%)	53 (42.7%)	11 (36.7%)	9 (56.3%)	33 (42.3%)	0.69
Electrolyte imbalance, *n* (%)	13 (10.5%)	2 (6.7%)	1 (6.3%)	10 (12.8%)	0.99
Acid base imbalance, *n* (%)	24 (19.4%)	6 (20.0%)	3 (18.8%)	15 (19.2%)	1.00
Uremic symptom, *n* (%)	1 (0.8%)	0 (0.0%)	0 (0.0%)	1 (1.3%)	0.21
Rhabdomyolysis, *n* (%)	5 (4.0%)	3 (10.0%)	0 (0.0%)	2 (2.6%)	0.57
Oliguria/anuria, *n* (%)	66 (53.2%)	18 (60.0%)	7 (43.8%)	41 (52.6%)	**0.001**
**Clinical parameters before initiating RRT (T2)**					
BUN, mg/dL	67.8 ± 42.3	76.2 ± 47.7	47.7 ± 37.2	68.7 ± 40.2	**0.003**
SCr, mg/dL	3.4 ± 2.3	3.6 ± 2.3	1.9 ± 1.4	3.6 ± 2.3	0.38
Potassium, mEq/L	4.4 ± 0.8	4.6 ± 0.9	4.2 ± 0.8	4.4 ± 0.8	**0.046**
Bicarbonate, mmol/L	19.4 ± 4.2	18.4 ± 3.8	17.7 ± 3.6	20.1 ± 4.3	0.84
SOFA score	10.7 ± 3.9	11.9 ± 3.7	12.7 ± 4.0	9.8 ± 3.7	0.22
qSOFA score	1.0 ± 0.8	1.1 ± 0.8	1.3 ± 0.9	0.9 ± 0.8	**0.048**
IE score	9.5 ± 12.6	11.6 ± 10.7	14.8 ± 19.9	7.6 ± 11.1	0.10
**Clinical parameters when weaning off RRT (T3)**					
SBP, mmHg	128.7 ± 24.2	129.4 ± 24.1	115.8 ± 24.6	131.1 ± 23.6	0.87
DBP, mmHg	68.3 ± 14.6	67.1 ± 15.7	69.1 ± 18.2	68.6 ± 13.5	**<0.001**
Body weight, kg	67.0 ± 14.4	65.6 ± 16.1	62.9 ± 16.5	68.4 ± 13.2	0.91
Urine output, ml	1030.3 ± 668.8	1003.9 ± 714.1	962.9 ± 516.5	1054.3 ± 684.7	0.07
Platelet count, **103/uL**	125.5 ± 74.2	128.6 ± 82.8	88.4 ± 57.9	131.9 ± 72.2	0.10
BUN, mg/dL	50.0 ± 23.2	48.7 ± 29.5	47.8 ± 20.4	50.9 ± 21.1	0.76
SCr, mg/dL	1.8 ± 1.6	1.7 ± 1.6	0.9 ± 0.3	2.0 ± 1.7	0.07
eGFR, mL/min/1.73 m^2^	62.8 ± 34.9	67.9 ± 36.9	104.3 ± 36.8	52.2 ± 26.2	**0.001**
Potassium, mEq/L	4.0 ± 0.6	4.0 ± 0.8	4.0 ± 0.7	4.0 ± 0.5	0.06
Bicarbonate, mmol/L	21.7 ± 3.8	20.7 ± 3.4	23.4 ± 3.1	21.8 ± 4.0	0.32
SOFA score	7.5 ± 2.9	7.1 ± 2.8	9.1 ± 3.1	7.4 ± 2.8	**0.015**
qSOFA score	0.8 ± 0.8	0.9 ± 0.8	1.3 ± 1.0	0.6 ± 0.7	0.59
uLFABP/Cr (log), μg/gCr	2.2 ± 0.7	2.7 ± 0.4	2.4 ± 0.8	2.0 ± 0.7	**<0.001**
uNGAL/Cr (log), μg/gCr	2.5 ± 0.6	2.8 ± 0.4	2.5 ± 0.8	2.3 ± 0.6	**<0.001**
**Clinical parameters after being weaned off RRT for 24 h (T4)**					
SCr, mg/dL	2.8 ± 1.8	2.5 ± 1.1	1.6 ± 0.8	3.2 ± 2.1	0.41
BUN, mg/dL	55.8 ± 24.6	56.9 ± 27.3	48.4 ± 21.5	56.9 ± 24.1	0.43
Daily UO (log), mL	3.1 ± 0.4	3.2 ± 0.3	3.2 ± 0.2	3.1 ± 0.4	0.71
**Outcome**					
Mortality, *n* (%)	38 (30.7%)	13 (43.3%)	12 (75.0%)	13 (16.7%)	**<0.001**
Mortality and re-dialysis, *n* (%)	63 (50.7%)	18 (60.0%)	12 (75.0%)	33 (42.3%)	**0.015**

**Abbreviations:** AKI, acute kidney injury; BUN, blood urea nitrogen; Cr, creatinine; DBP, diastolic blood pressure; eGFR, estimated glomerular filtration rate; IE, inotrope exposure; qSOFA, quick Sequential Organ Failure Assessment; RRT, renal replacement therapy; SBP, systolic blood pressure; SCr, serum creatinine; SOFA, Sequential Organ Failure Assessment; uL-FABP, urinary liver-type fatty acid binding protein; uNGAL, urinary neutrophil gelatinase-associated lipocalin; UO, urine output. Differences between groups were assessed by a one-way ANOVA with post-hoc Bonferroni tests for multiple comparisons and contrast analysis. * Nephrotoxin, pigment nephropathy, contrast nephropathy, hepatorenal syndrome

**Table 2 biomedicines-10-01628-t002:** Cox proportional hazards models depicting the 90-day mortality.

Parameter	Hazard Ratio	95% Confidence Interval	*p*-Value
**Demographic factors (T1)**			
Age, year	1.06	1.02–1.10	**0.001**
Gender, *n* (%)	1.26	0.53–2.96	0.60
Diabetic mellitus, *n* (%)	1.55	0.63–3.83	0.35
Baseline eGFR, mL/min/1.73 m^2^	1.02	1.00–1.05	**0.049**
**Indication for dialysis**			
Azotemia, *n* (%)	0.56	0.20–1.64	0.29
Fluid overload, *n* (%)	0.66	0.24–1.82	0.42
Electrolyte imbalance, *n* (%)	0.62	0.12–3.38	0.59
Acid-base imbalance, *n* (%)	2.59	1.12–5.98	**0.026**
Rhabdomyolysis, *n* (%)	2.98	0.28–31.93	0.37
Oliguria/anuria, *n* (%)	0.47	0.17–1.25	0.13
**Clinical parameters before initiating RRT** **(T2)**			
BUN, mg/dL	1.01	1.00–1.02	**0.045**
SOFA score	1.07	0.94–1.22	0.31
**Clinical parameters when weaning off RRT (T3)**			
SBP, mmHg	0.99	0.97–1.01	0.26
Body weight, kg	0.99	0.96–1.02	0.44
Daily UO (log), ml	0.69	0.16–2.91	0.61
BUN, mg/dL	0.99	0.97–1.01	0.53
eGFR, mL/min/1.73 m^2^	0.96	0.94–0.99	**0.01**
Potassium, mEq/L	1.33	0.66–2.66	0.43
SOFA	1.10	0.93–1.30	0.25
uNGAL/Cr (log), μg/gCr	3.68	1.63–8.31	**0.002**
**Clinical parameters after being weaned off RRT for 24 h (T4)**			
Daily UO (log), ml	2.46	0.45–13.57	0.30
BUN, mg/dL	1.01	0.99–1.024	0.58
Cluster 1 vs. 3	3.69	1.25–10.93	**0.018**
Cluster 2 vs. 3	26.19	5.42–126.65	**<0.001**

**Abbreviations:** AKI, acute kidney injury; BUN, blood urea nitrogen; Cr, creatinine; eGFR, estimated glomerular filtration rate; RRT, renal replacement therapy; SBP, systolic blood pressure; SOFA, Sequential Organ Failure Assessment; uNGAL, urinary neutrophil gelatinase-associated lipocalin; UO, urine output.

**Table 3 biomedicines-10-01628-t003:** Cox proportional hazards models depicting the composite outcome of 90-day mortality or re-dialysis.

Parameter	Hazard Ratio	95% Confidence Interval	*p*-Value
**Demographic factors (T1)**			
Age, year	1.03	1.00–1.10	**0.003**
Gender, *n* (%)	1.41	0.66–3.01	0.38
Diabetic mellitus, *n* (%)	1.44	0.71–2.92	0.31
Baseline eGFR, mL/min/1.73 m^2^	1.00	0.98–1.02	0.89
**Indication for dialysis**			
Azotemia, *n* (%)	0.64	0.30–1.36	0.25
Fluid overload, *n* (%)	1.06	0.54–2.11	0.87
Electrolyte imbalance, *n* (%)	0.79	0.25–2.52	0.69
Acid- base imbalance, *n* (%)	1.98	0.97–4.06	0.06
Rhabdomyolysis, *n* (%)	3.12	0.69–14.21	0.14
Oliguria/anuria, *n* (%)	0.82	0.39–1.71	0.60
**Clinical parameters before initiating RRT** **(T2)**			
BUN, mg/dL	1.01	1.00–1.02	0.23
SOFA score	0.99	0.89–1.10	0.87
**Clinical parameters when weaning off RRT (T3)**			
SBP, mmHg	1.00	0.99–1.02	0.85
Body weight, kg	0.99	0.97–1.02	0.51
Daily UO (log), ml	1.75	0.63–4.86	0.29
BUN, mg/dL	1.00	0.99–1.02	0.59
eGFR, mL/min/1.73 m^2^	0.98	0.96–1.00	0.05
Potassium, mEq/L	1.32	0.76–2.30	0.33
SOFA	1.08	0.95–1.23	0.24
uNGAL/Cr (log), μg/gCr	2.43	1.36–4.33	**0.003**
**Clinical parameters after being weaned off RRT for 24 h (T4)**			
Daily UO (log), mL	0.85	0.32–2.27	0.75
BUN, mg/dL	1.00	0.99–1.02	0.64
Cluster 1 vs. 3	2.70	1.11–6.57	**0.028**
Cluster 2 vs. 3	44.53	11.92–166.39	**<0.001**

**Abbreviations:** AKI, acute kidney injury; BUN, blood urea nitrogen; Cr, creatinine; eGFR, estimated glomerular filtration rate; RRT, renal replacement therapy; SBP, systolic blood pressure; SOFA, Sequential Organ Failure Assessment; uNGAL, urinary neutrophil gelatinase-associated lipocalin; UO, urine output.

## Data Availability

The data presented in this study are available on request from the corresponding author. The data are not publicly available due to ethical and privacy restrictions.

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
