# Peer review of "Distinct Subtyping of Successful Weaning from Acute Kidney Injury Requiring Renal Replacement Therapy by Consensus Clustering in Critically Ill Patients"

_biomedicines, 2022, doi:10.3390/biomedicines10071628_

Round 1

Reviewer 1 Report

The study was aimed to create a multifactorial scoring system, suitable for better prediction of outcome in critically ill patients with AKI, treated with RRT. The analysis was made with use of very sophisticated combination of clinical and laboratory makers.  Nevertheless that the authors were able to demonstrate some correlations, the overall impression is that this model is too complicated to be used in clinical practice. There are no data on several basic parameters, such as a number of  used  catecholamines, incidence of mechanical ventilation, suspected the triggers (causes) of the  sepsis and shock the (predominant clinical issues) , timing from diagnosis  to RRT, use of not use of nephrotoxic antibiotics. Some clear association seems to be between the highest SOFA score in patients from cluster 2 and the worst outcome. Looks like the extra-renal or multiorgan issues determined the outcome in patients from cluster 2, however it was not clarified among several parameters and nevertheless use of  a complicated statistical model.

NGAL is just a marker of tubular injury, secondary to other casues, and cannot be indicated as a marker predicting  mortality, even though there is some statistical correlation.

Such a study with hard workload (which is highly  apppreciated), in terms of data collection and statistical analysis – should be concluded with some clear, straightforward message for  relevant clinicians.

The conclusion here „ The consensus clustering algorithm identified three distinct clusters of critically ill patients with AKI-RRT with different outcomes using demographic and clinical variables and biomarkers. Urinary NGAL/Cr and distinct clustering phenotypes could predict 90-day mortality or re-dialysis” is a summary of statistical analysis, not a clinical message, nor conclusion.

Please revise the discussion and indicate real significant clinical factors important for clinicians decisions.  The balance towards „More clinics – less statistics”  will improve the quality of the report.

Minor comments:

1.      Please clarify criterion 3 „ the indication for starting RRT was remission”

„We enrolled critically ill adult patients with AKI-RRT who met the following criteria: (1) those whose intrinsic renal function had adequately recovered 24; (2) RRT was no longer consistent with the treatment goals 24; (3) the indication for starting RRT was remission; and (4) a trend toward decreasing serum creatinine (sCr), urine output ≥ 400 mL/24 hours with or without diuretics, and improved fluid overload, electrolyte and metabolic status”

2.      The patients defined as „cluster 2” presented the worst prognosis, despite the best pre-dialysis renal function and other clinical factors were in charge for the  overall poor outcome. Please add more details and  specific data on pre-existing comorbidities (multiorgan failure?), which could have impact on this association.

The study was aimed to create a multifactorial scoring system, suitable for better prediction of outcome in critically ill patients with AKI, treated with RRT. The analysis was made with use of very sophisticated combination of clinical and laboratory makers.  Nevertheless that the authors were able to demonstrate some correlations, the overall impression is that this model is too complicated to be used in clinical practice. There are no data on several basic parameters, such as a number of  used  catecholamines, incidence of mechanical ventilation, suspected the triggers (causes) of the  sepsis and shock the (predominant clinical issues) , timing from diagnosis  to RRT, use of not use of nephrotoxic antibiotics. Some clear association seems to be between the highest SOFA score in patients from cluster 2 and the worst outcome. Looks like the extra-renal or multiorgan issues determined the outcome in patients from cluster 2, however it was not clarified among several parameters and nevertheless use of  a complicated statistical model.

NGAL is just a marker of tubular injury, secondary to other casues, and cannot be indicated as a marker predicting  mortality, even though there is some statistical correlation.

Such a study with hard workload (which is highly  apppreciated), in terms of data collection and statistical analysis – should be concluded with some clear, straightforward message for  relevant clinicians.

The conclusion here „ The consensus clustering algorithm identified three distinct clusters of critically ill patients with AKI-RRT with different outcomes using demographic and clinical variables and biomarkers. Urinary NGAL/Cr and distinct clustering phenotypes could predict 90-day mortality or re-dialysis” is a summary of statistical analysis, not a clinical message, nor conclusion.

Please revise the discussion and indicate real significant clinical factors important for clinicians decisions.  The balance towards „More clinics – less statistics”  will improve the quality of the report.

Minor comments:

1.      Please clarify criterion 3 „ the indication for starting RRT was remission”

„We enrolled critically ill adult patients with AKI-RRT who met the following criteria: (1) those whose intrinsic renal function had adequately recovered 24; (2) RRT was no longer consistent with the treatment goals 24; (3) the indication for starting RRT was remission; and (4) a trend toward decreasing serum creatinine (sCr), urine output ≥ 400 mL/24 hours with or without diuretics, and improved fluid overload, electrolyte and metabolic status”

2.      The patients defined as „cluster 2” presented the worst prognosis, despite the best pre-dialysis renal function and other clinical factors were in charge for the  overall poor outcome. Please add more details and  specific data on pre-existing comorbidities (multiorgan failure?), which could have impact on this association.

The study was aimed to create a multifactorial scoring system, suitable for better prediction of outcome in critically ill patients with AKI, treated with RRT. The analysis was made with use of very sophisticated combination of clinical and laboratory makers.  Nevertheless that the authors were able to demonstrate some correlations, the overall impression is that this model is too complicated to be used in clinical practice. There are no data on several basic parameters, such as a number of  used  catecholamines, incidence of mechanical ventilation, suspected the triggers (causes) of the  sepsis and shock the (predominant clinical issues) , timing from diagnosis  to RRT, use of not use of nephrotoxic antibiotics. Some clear association seems to be between the highest SOFA score in patients from cluster 2 and the worst outcome. Looks like the extra-renal or multiorgan issues determined the outcome in patients from cluster 2, however it was not clarified among several parameters and nevertheless use of  a complicated statistical model.

NGAL is just a marker of tubular injury, secondary to other casues, and cannot be indicated as a marker predicting  mortality, even though there is some statistical correlation.

Such a study with hard workload (which is highly  apppreciated), in terms of data collection and statistical analysis – should be concluded with some clear, straightforward message for  relevant clinicians.

The conclusion here „ The consensus clustering algorithm identified three distinct clusters of critically ill patients with AKI-RRT with different outcomes using demographic and clinical variables and biomarkers. Urinary NGAL/Cr and distinct clustering phenotypes could predict 90-day mortality or re-dialysis” is a summary of statistical analysis, not a clinical message, nor conclusion.

Please revise the discussion and indicate real significant clinical factors important for clinicians decisions.  The balance towards „More clinics – less statistics”  will improve the quality of the report.

Minor comments:

1.      Please clarify criterion 3 „ the indication for starting RRT was remission”

„We enrolled critically ill adult patients with AKI-RRT who met the following criteria: (1) those whose intrinsic renal function had adequately recovered 24; (2) RRT was no longer consistent with the treatment goals 24; (3) the indication for starting RRT was remission; and (4) a trend toward decreasing serum creatinine (sCr), urine output ≥ 400 mL/24 hours with or without diuretics, and improved fluid overload, electrolyte and metabolic status”

2.      The patients defined as „cluster 2” presented the worst prognosis, despite the best pre-dialysis renal function and other clinical factors were in charge for the  overall poor outcome. Please add more details and  specific data on pre-existing comorbidities (multiorgan failure?), which could have impact on this association.

The study was aimed to create a multifactorial scoring system, suitable for better prediction of outcome in critically ill patients with AKI, treated with RRT. The analysis was made with use of very sophisticated combination of clinical and laboratory makers.  Nevertheless that the authors were able to demonstrate some correlations, the overall impression is that this model is too complicated to be used in clinical practice. There are no data on several basic parameters, such as a number of  used  catecholamines, incidence of mechanical ventilation, suspected the triggers (causes) of the  sepsis and shock the (predominant clinical issues) , timing from diagnosis  to RRT, use of not use of nephrotoxic antibiotics. Some clear association seems to be between the highest SOFA score in patients from cluster 2 and the worst outcome. Looks like the extra-renal or multiorgan issues determined the outcome in patients from cluster 2, however it was not clarified among several parameters and nevertheless use of  a complicated statistical model.

NGAL is just a marker of tubular injury, secondary to other casues, and cannot be indicated as a marker predicting  mortality, even though there is some statistical correlation.

Such a study with hard workload (which is highly  apppreciated), in terms of data collection and statistical analysis – should be concluded with some clear, straightforward message for  relevant clinicians.

The conclusion here „ The consensus clustering algorithm identified three distinct clusters of critically ill patients with AKI-RRT with different outcomes using demographic and clinical variables and biomarkers. Urinary NGAL/Cr and distinct clustering phenotypes could predict 90-day mortality or re-dialysis” is a summary of statistical analysis, not a clinical message, nor conclusion.

Please revise the discussion and indicate real significant clinical factors important for clinicians decisions.  The balance towards „More clinics – less statistics”  will improve the quality of the report.

Minor comments:

1.      Please clarify criterion 3 „ the indication for starting RRT was remission”

„We enrolled critically ill adult patients with AKI-RRT who met the following criteria: (1) those whose intrinsic renal function had adequately recovered 24; (2) RRT was no longer consistent with the treatment goals 24; (3) the indication for starting RRT was remission; and (4) a trend toward decreasing serum creatinine (sCr), urine output ≥ 400 mL/24 hours with or without diuretics, and improved fluid overload, electrolyte and metabolic status”

2.      The patients defined as „cluster 2” presented the worst prognosis, despite the best pre-dialysis renal function and other clinical factors were in charge for the  overall poor outcome. Please add more details and  specific data on pre-existing comorbidities (multiorgan failure?), which could have impact on this association.

The study was aimed to create a multifactorial scoring system, suitable for better prediction of outcome in critically ill patients with AKI, treated with RRT. The analysis was made with use of very sophisticated combination of clinical and laboratory makers.  Nevertheless that the authors were able to demonstrate some correlations, the overall impression is that this model is too complicated to be used in clinical practice. There are no data on several basic parameters, such as a number of  used  catecholamines, incidence of mechanical ventilation, suspected the triggers (causes) of the  sepsis and shock the (predominant clinical issues) , timing from diagnosis  to RRT, use of not use of nephrotoxic antibiotics. Some clear association seems to be between the highest SOFA score in patients from cluster 2 and the worst outcome. Looks like the extra-renal or multiorgan issues determined the outcome in patients from cluster 2, however it was not clarified among several parameters and nevertheless use of  a complicated statistical model.

NGAL is just a marker of tubular injury, secondary to other casues, and cannot be indicated as a marker predicting  mortality, even though there is some statistical correlation.

Such a study with hard workload (which is highly  apppreciated), in terms of data collection and statistical analysis – should be concluded with some clear, straightforward message for  relevant clinicians.

The conclusion here „ The consensus clustering algorithm identified three distinct clusters of critically ill patients with AKI-RRT with different outcomes using demographic and clinical variables and biomarkers. Urinary NGAL/Cr and distinct clustering phenotypes could predict 90-day mortality or re-dialysis” is a summary of statistical analysis, not a clinical message, nor conclusion.

Please revise the discussion and indicate real significant clinical factors important for clinicians decisions.  The balance towards „More clinics – less statistics”  will improve the quality of the report.

Minor comments:

1.      Please clarify criterion 3 „ the indication for starting RRT was remission”

„We enrolled critically ill adult patients with AKI-RRT who met the following criteria: (1) those whose intrinsic renal function had adequately recovered 24; (2) RRT was no longer consistent with the treatment goals 24; (3) the indication for starting RRT was remission; and (4) a trend toward decreasing serum creatinine (sCr), urine output ≥ 400 mL/24 hours with or without diuretics, and improved fluid overload, electrolyte and metabolic status”

2.      The patients defined as „cluster 2” presented the worst prognosis, despite the best pre-dialysis renal function and other clinical factors were in charge for the  overall poor outcome. Please add more details and  specific data on pre-existing comorbidities (multiorgan failure?), which could have impact on this association.

The study was aimed to create a multifactorial scoring system, suitable for better prediction of outcome in critically ill patients with AKI, treated with RRT. The analysis was made with use of very sophisticated combination of clinical and laboratory makers.  Nevertheless that the authors were able to demonstrate some correlations, the overall impression is that this model is too complicated to be used in clinical practice. There are no data on several basic parameters, such as a number of  used  catecholamines, incidence of mechanical ventilation, suspected the triggers (causes) of the  sepsis and shock the (predominant clinical issues) , timing from diagnosis  to RRT, use of not use of nephrotoxic antibiotics. Some clear association seems to be between the highest SOFA score in patients from cluster 2 and the worst outcome. Looks like the extra-renal or multiorgan issues determined the outcome in patients from cluster 2, however it was not clarified among several parameters and nevertheless use of  a complicated statistical model.

NGAL is just a marker of tubular injury, secondary to other casues, and cannot be indicated as a marker predicting  mortality, even though there is some statistical correlation.

Such a study with hard workload (which is highly  apppreciated), in terms of data collection and statistical analysis – should be concluded with some clear, straightforward message for  relevant clinicians.

The conclusion here „ The consensus clustering algorithm identified three distinct clusters of critically ill patients with AKI-RRT with different outcomes using demographic and clinical variables and biomarkers. Urinary NGAL/Cr and distinct clustering phenotypes could predict 90-day mortality or re-dialysis” is a summary of statistical analysis, not a clinical message, nor conclusion.

Please revise the discussion and indicate real significant clinical factors important for clinicians decisions.  The balance towards „More clinics – less statistics”  will improve the quality of the report.

Minor comments:

1.      Please clarify criterion 3 „ the indication for starting RRT was remission”

„We enrolled critically ill adult patients with AKI-RRT who met the following criteria: (1) those whose intrinsic renal function had adequately recovered 24; (2) RRT was no longer consistent with the treatment goals 24; (3) the indication for starting RRT was remission; and (4) a trend toward decreasing serum creatinine (sCr), urine output ≥ 400 mL/24 hours with or without diuretics, and improved fluid overload, electrolyte and metabolic status”

2.      The patients defined as „cluster 2” presented the worst prognosis, despite the best pre-dialysis renal function and other clinical factors were in charge for the  overall poor outcome. Please add more details and  specific data on pre-existing comorbidities (multiorgan failure?), which could have impact on this association.

The study was aimed to create a multifactorial scoring system, suitable for better prediction of outcome in critically ill patients with AKI, treated with RRT. The analysis was made with use of very sophisticated combination of clinical and laboratory makers.  Nevertheless that the authors were able to demonstrate some correlations, the overall impression is that this model is too complicated to be used in clinical practice. There are no data on several basic parameters, such as a number of  used  catecholamines, incidence of mechanical ventilation, suspected the triggers (causes) of the  sepsis and shock the (predominant clinical issues) , timing from diagnosis  to RRT, use of not use of nephrotoxic antibiotics. Some clear association seems to be between the highest SOFA score in patients from cluster 2 and the worst outcome. Looks like the extra-renal or multiorgan issues determined the outcome in patients from cluster 2, however it was not clarified among several parameters and nevertheless use of  a complicated statistical model.

NGAL is just a marker of tubular injury, secondary to other casues, and cannot be indicated as a marker predicting  mortality, even though there is some statistical correlation.

Such a study with hard workload (which is highly  apppreciated), in terms of data collection and statistical analysis – should be concluded with some clear, straightforward message for  relevant clinicians.

The conclusion here „ The consensus clustering algorithm identified three distinct clusters of critically ill patients with AKI-RRT with different outcomes using demographic and clinical variables and biomarkers. Urinary NGAL/Cr and distinct clustering phenotypes could predict 90-day mortality or re-dialysis” is a summary of statistical analysis, not a clinical message, nor conclusion.

Please revise the discussion and indicate real significant clinical factors important for clinicians decisions.  The balance towards „More clinics – less statistics”  will improve the quality of the report.

Minor comments:

1.      Please clarify criterion 3 „ the indication for starting RRT was remission”

„We enrolled critically ill adult patients with AKI-RRT who met the following criteria: (1) those whose intrinsic renal function had adequately recovered 24; (2) RRT was no longer consistent with the treatment goals 24; (3) the indication for starting RRT was remission; and (4) a trend toward decreasing serum creatinine (sCr), urine output ≥ 400 mL/24 hours with or without diuretics, and improved fluid overload, electrolyte and metabolic status”

2.      The patients defined as „cluster 2” presented the worst prognosis, despite the best pre-dialysis renal function and other clinical factors were in charge for the  overall poor outcome. Please add more details and  specific data on pre-existing comorbidities (multiorgan failure?), which could have impact on this association.

The study was aimed to create a multifactorial scoring system, suitable for better prediction of outcome in critically ill patients with AKI, treated with RRT. The analysis was made with use of very sophisticated combination of clinical and laboratory makers.  Nevertheless that the authors were able to demonstrate some correlations, the overall impression is that this model is too complicated to be used in clinical practice. There are no data on several basic parameters, such as a number of  used  catecholamines, incidence of mechanical ventilation, suspected the triggers (causes) of the  sepsis and shock the (predominant clinical issues) , timing from diagnosis  to RRT, use of not use of nephrotoxic antibiotics. Some clear association seems to be between the highest SOFA score in patients from cluster 2 and the worst outcome. Looks like the extra-renal or multiorgan issues determined the outcome in patients from cluster 2, however it was not clarified among several parameters and nevertheless use of  a complicated statistical model.

NGAL is just a marker of tubular injury, secondary to other casues, and cannot be indicated as a marker predicting  mortality, even though there is some statistical correlation.

Such a study with hard workload (which is highly  apppreciated), in terms of data collection and statistical analysis – should be concluded with some clear, straightforward message for  relevant clinicians.

The conclusion here „ The consensus clustering algorithm identified three distinct clusters of critically ill patients with AKI-RRT with different outcomes using demographic and clinical variables and biomarkers. Urinary NGAL/Cr and distinct clustering phenotypes could predict 90-day mortality or re-dialysis” is a summary of statistical analysis, not a clinical message, nor conclusion.

Please revise the discussion and indicate real significant clinical factors important for clinicians decisions.  The balance towards „More clinics – less statistics”  will improve the quality of the report.

Minor comments:

1.      Please clarify criterion 3 „ the indication for starting RRT was remission”

„We enrolled critically ill adult patients with AKI-RRT who met the following criteria: (1) those whose intrinsic renal function had adequately recovered 24; (2) RRT was no longer consistent with the treatment goals 24; (3) the indication for starting RRT was remission; and (4) a trend toward decreasing serum creatinine (sCr), urine output ≥ 400 mL/24 hours with or without diuretics, and improved fluid overload, electrolyte and metabolic status”

2.      The patients defined as „cluster 2” presented the worst prognosis, despite the best pre-dialysis renal function and other clinical factors were in charge for the  overall poor outcome. Please add more details and  specific data on pre-existing comorbidities (multiorgan failure?), which could have impact on this association.

Author Response

Reviewer #1:

The study was aimed to create a multifactorial scoring system, suitable for better prediction of outcome in critically ill patients with AKI, treated with RRT. The analysis was made with use of very sophisticated combination of clinical and laboratory makers. Nevertheless that the authors were able to demonstrate some correlations, the overall impression is that this model is too complicated to be used in clinical practice. There are no data on several basic parameters, such as a number of used catecholamines, incidence of mechanical ventilation, suspected the triggers (causes) of the sepsis and shock the (predominant clinical issues) , timing from diagnosis to RRT, use of not use of nephrotoxic antibiotics. Some clear association seems to be between the highest SOFA score in patients from cluster 2 and the worst outcome. Looks like the extra-renal or multiorgan issues determined the outcome in patients from cluster 2, however it was not clarified among several parameters and nevertheless use of a complicated statistical model.

Response: Thank you for your comment. We have also added IE score before receiving RRT, as well as the incidence of infection, nephrotoxic agents induced AKI and contrast induced AKI in the revised table 1. The incidence of mechanical ventilation had been reported in the original table 1. While it is not perfect, our work is the first study to identify the possibility to predict 90-day mortality or re-dialysis. We have added some associated description and revised the Discussion section (P.35). In critical settings, multifactorial conditions could relate to sepsis or shock related AKI. It is challenge to identify the etiologies of AKI, however the main purpose of our study was on the indicators for patients who could wean from dialysis requiring AKI. We raised the importance of uNGAL/Cr, a marker of kidney injury was associated with successful weaning from dialysis requiring AKI. Although some of the important information we could not provide, as your comments, we have acknowledged them in the study limitations.

NGAL is just a marker of tubular injury, secondary to other casues, and cannot be indicated as a marker predicting mortality, even though there is some statistical correlation.

Response: Thank you for your comment. In the literature, NGAL levels have been a useful prognostic tool for predicting acute kidney injury and mortality [1-3]. In regard to your comments, we changed “independent predictor” and “predict” and to “independent factor” (P.25 and P.34) and “be associated with” (P.32), respectively.

[Reference]

  1. Haase, M.; Bellomo, R.;  Devarajan, P.;  Schlattmann, P.;  Haase-Fielitz, A.; Group, N. M.-a. I., Accuracy of neutrophil gelatinase-associated lipocalin (NGAL) in diagnosis and prognosis in acute kidney injury: a systematic review and meta-analysis. American journal of kidney diseases 2009, 54 (6), 1012-1024.
  2. Helmersson-Karlqvist, J.; Larsson, A.;  Carlsson, A. C.;  Venge, P.;  Sundström, J.;  Ingelsson, E.;  Lind, L.; Ärnlöv, J., Urinary neutrophil gelatinase-associated lipocalin (NGAL) is associated with mortality in a community-based cohort of older Swedish men. Atherosclerosis 2013, 227 (2), 408-413.
  3. Mahmoodpoor, A.; Hamishehkar, H.;  Fattahi, V.;  Sanaie, S.;  Arora, P.; Nader, N. D., Urinary versus plasma neutrophil gelatinase-associated lipocalin (NGAL) as a predictor of mortality for acute kidney injury in intensive care unit patients. Journal of clinical anesthesia 2018, 44, 12-17.

Such a study with hard workload (which is highly appreciated), in terms of data collection and statistical analysis – _should be concluded with some clear, straightforward message for relevant clinicians.

Response: Thanks for your appreciation. We have revised the Discussion section. We have appraised the sentence and reword the conclusion as “ In summary, the three clusters of AKI-RRT patients who attempted to wean off RRT had discrete features and were highly associated with mortality and re-dialysis“(P.37).

The conclusion here „ _The consensus clustering algorithm identified three distinct clusters of critically ill patients with AKI-RRT with different outcomes using demographic and clinical variables and biomarkers. Urinary NGAL/Cr and distinct clustering phenotypes could predict 90-day mortality or re-dialysis” _is a summary of statistical analysis, not a clinical message, nor conclusion.

Response: Thank you for your comment. We have revised the conclusion to be more directly informative in the Abstract section (P.4).

Please revise the discussion and indicate real significant clinical factors important for clinicians decisions. The balance towards „More clinics – _less statistics” _will improve the quality of the report.

Response: Thank you for your comment. We have revised the Discussion section (P.31, P.33).

Minor comments:

  1. Please clarify criterion 3 „the indication for starting RRT was remission”

„We enrolled critically ill adult patients with AKI-RRT who met the following criteria: (1) those whose intrinsic renal function had adequately recovered; (2) RRT was no longer consistent with the treatment goals; (3) the indication for starting RRT was remission; and (4) a trend toward decreasing serum creatinine (sCr), urine output ≥ 400 mL/24 hours with or without diuretics, and improved fluid overload, electrolyte and metabolic status”

Response: Thank you for your comment. The indication of starting RRT including azotemia with overt uremic symptoms, refractory hyperkalemia, oliguria or anuria (< 100 ml/24 h) refractory to diuretics, fluid overload refractory to diuretics along with a central venous pressure > 12 mmHg or pulmonary edema with PaO2/FiO2 < 300 mmHg, as well as severe metabolic acidosis. We have revised the Method section (P.7).

  1. The patients defined as „cluster 2” presented the worst prognosis, despite the best pre-dialysis renal function and other clinical factors were in charge for the overall poor outcome. Please add more details and specific data on pre-existing comorbidities (multiorgan failure?), which could have impact on this association.

Response: Thank you for your suggestion. Information on pre-existing comorbidities was listed in table 1. We would like to point out the differences in SOFA scores and qSOFA scores among the three clusters were insignificant and borderline significant, respectively. This may indicate that baseline organ functions were not significantly different among the three clusters. We have added some associated description and revised the Discussion section (P.35).

Reviewer 2 Report

In this retrospective study of ARI patients who underwent renal replacement therapy (RRT), consensus grouping algorithms based on 45 parameters were used to find different phenotypes. These subgroup data could help to determine optimal time of RTT withdrawal, as well as to assume the prognosis of these critically ill patients.

Author Response

Reviewer #2:

In this retrospective study of ARI patients who underwent renal replacement therapy (RRT), consensus grouping algorithms based on 45 parameters were used to find different phenotypes. These subgroup data could help to determine optimal time of RTT withdrawal, as well as to assume the prognosis of these critically ill patients.

Response: Thanks for your appreciation. 

Round 2

Reviewer 1 Report

The revised version includes relevant feedback to the previous comments and suggestions.  No further comments.